# Recent Trends and Developments in the Electrical Discharge Machining Industry: A Review

Anna A. Kamenskikh [1], Karim R. Muratov [2], Evgeny S. Shlykov [2], Sarabjeet Singh Sidhu [3,*], Amit Mahajan [4], Yulia S. Kuznetsova [1] and Timur R. Ablyaz [2,*]

1. Department of Computational Mathematics, Mechanics and Biomechanics, Perm National Research Polytechnic University, 614990 Perm, Russia; anna_kamenskih@mail.ru (A.A.K.); suhodolchik@mail.ru (Y.S.K.)
2. Department of Mechanical Engineering, Perm National Research Polytechnic University, 614000 Perm, Russia; karimur_80@mail.ru (K.R.M.); kruspert@mail.ru (E.S.S.)
3. Mechanical Engineering Department, Sardar Beant Singh State University, Gurdaspur 143521, India
4. Department of Mechanical Engineering, Khalsa College of Engineering & Technology, Amritsar 143001, India; amitmahajan291@gmail.com

* Correspondence: sarabjeetsidhu@yahoo.com (S.S.S.); lowrider11-13-11@mail.ru (T.R.A.)

**Abstract:** Electrical discharge machining (EDM) is a highly precise technology that not only facilitates the machining of components into desired shapes but also enables the alteration of the physical and chemical properties of workpieces. The complexity of the process is due to a number of regulating factors such as the material of the workpiece and tools, dielectric medium, and other process parameters. Based on the material type, electrode shape, and process configuration, various shapes and degrees of accuracy can be generated. The study of erosion is based on research into processing techniques, which are the primary tools for using EDM. Empirical knowledge with subsequent optimization of technological parameters is one of the ways to obtain the required surface quality of the workpiece with defect minimization, as well as mathematical and numerical modeling of the EDM process. This article critically examines all key aspects of EDM, reflecting both the early foundations of electrical erosion and the current state of the industry, noting the current trends towards the transition of EDM to the 5.0 industry zone in terms of safety and minimizing the impact of the process on the environment.

**Keywords:** EDM; workpiece; electrode tool (ET); dielectric; technology; statistics; optimization; modeling





## 1. Introduction

At the outset of his scientific career in the 1760s, the English chemist Joseph Priestley embarked on a journey both in England and America supported by fellow scientists John Canton, William Watson, Timothy Lane, and Benjamin Franklin to study the history of the emergence and development of electricity [1]. Their study of existing experimental results and the influence of Benjamin Franklin catalyzed their own innovative and original research [2]. One of the most interesting results of the experiments of J. Priestly was an observation of the metal erosion effect under the action of an electric charge [3,4]. With further research, the scientist abandoned the topic of electricity and became more famous for his scientific research in pneumatic chemistry. However, the effect of material erosion with electric discharge formed the basis of non-traditional technological methods. Electric arc welding, proposed in the 1800s by Vasily Petrov, remained the main method of shaping using electric discharges for quite a long time [5]. As stated in 1938 by L.A. Yutkin, the electrohydraulic effect [6] became the motivation for the development of the method of electro-spark processing of metals, founded by Boris and Natalia Lazarenko [7]. On 3 April 1943, the priority in the discovery of a fundamentally new method of metal

processing was confirmed by these USSR authors. In 1946, the electro-erosion method for processing conductive materials was patented in five more countries, namely France, USA, Switzerland, Great Britain, and Sweden [8]. The scientific foundations of electroerosive machining and the main stages of development are reflected in a monograph [7]. However, the technology was widely developed after the introduction of a numerical control (NC) system to increase the accuracy of the process [9]. The studies of electro-erosive piercing were initially aimed at optimization, control, and planning of the process, as well as theoretical description and modeling of thermophysical analysis [10]. In 1989 and 1993, the first attempts were made to mathematically describe the material removal process in terms of the cathode, anode, and plasma [11–13]. Even later, the models of Dibitonto et al., based on the electrothermal concept of process description, contend with several modern studies regarding the description of the material removal rate (MRR) ratio [14]. According to B. Lazarenko, the reign of mechanical metal processing is over and electrical forces will take its place as a more highly organized process and the future belongs to a new material processing. Now electrical discharge machining (EDM) can be safely called one of the greatest breakthroughs of the 20th century. There are many studies aimed at reviewing the technological process and the trajectories of its development, classifying the main areas of research on EDM and existing machines, as well as the current state of the problem from various points of view [15–24]. At the same time, most studies consider new developments from the components of the process such as experimentation, optimization, processing of individual materials, surface morphology, modeling, application in science and technology, micro-electro erosive machining, surface modification, ecology, etc. [25–32]. Most review articles present in the literature deal with technology and the process experimental component. This article is based on the extensive research experience on this technology by the "Additive Technologies Center of Perm National Research Polytechnic University (Perm, Russia)". The area of interest of the scientific team is developing technology, experimentation, and modeling techniques which will make it possible to make a complete comprehensive analysis of the current state of the EDM industry. The main sections of this article are shown in Figure 1.

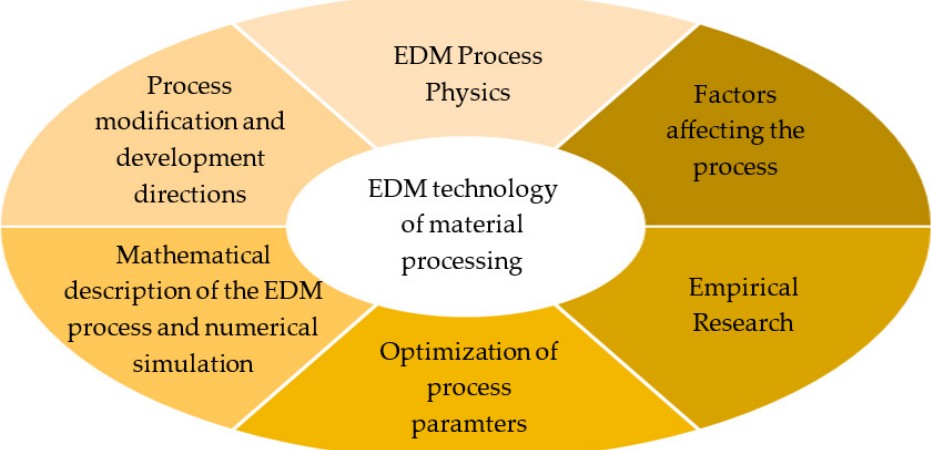

**Figure 1.** The main aspects of EDM that are discussed in this article.

## 2. EDM Processing

Pulsed electrophysical action involves the application of highly concentrated electrical energy to alter the shape, size, roughness, and surface properties in EDM. Figure 2 illustrates the general schematic of this process.

Initially, this technology found its application in shaping the surfaces of conductive hard-to-machine materials with complex physical and mechanical properties [7]. Over time, the scope of this technology expanded to include materials with low electrical conductivity, such as ceramics and composites [29,33]. The fundamental principle of

this technology relies on non-contact material removal from both the cathode and anode in a dielectric medium, achieved through the creation of a plasma channel generated by a highly concentrated electric discharge within a specific gap [34]. The energy from the discharge transforms into thermal energy, reaching temperatures between 6000 and 12,000 °C. In this superheated state, the material melts and evaporates, and the resulting molten material is washed away by the dielectric liquid, forming craters in a phenomenon known as electrical erosion [35,36].

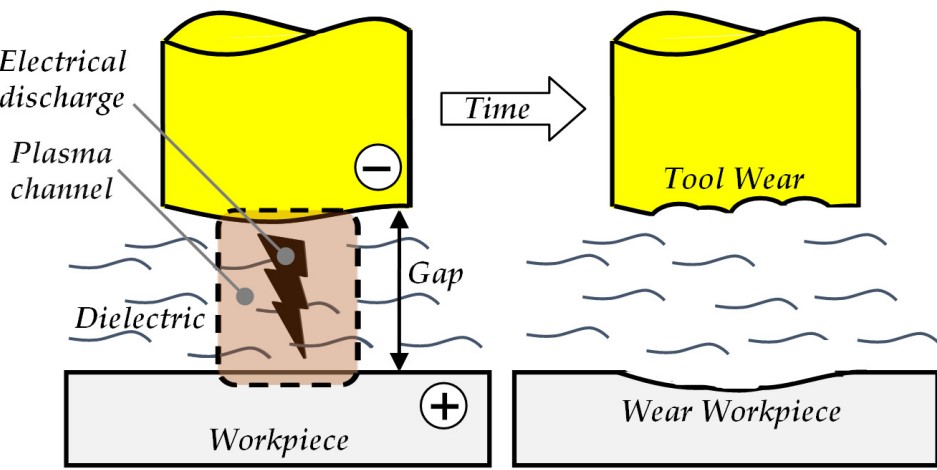

**Figure 2.** Schematic represents the basic principle and working of the EDM technology.

The final outcome is influenced by various factors, including the operational modes and types of the technological process, the processing mechanism, the experimental setup, the choice of dielectric, the electrode tool (ET), the workpiece material, and the condition of both the cathode and anode surfaces [25]. Depending on machining efficiency and surface layer requirements, the process is categorized into four types: roughing [37], semi-finishing [38], finishing [39], and super-finishing [40]. Roughing, or pre-treatment, prioritizes high productivity but may sacrifice surface quality. Semi-finishing aims for a smooth surface free from defects like peaks and valleys, scratches, and chips. Finishing results in a surface without technological allowances, often with the desired shape, dimensions, and required roughness. Superfinishing is employed to create specific micro-level surface structures or relief patterns. Achieving a particular type of surface treatment is directly tied to productivity and necessitates specialized equipment and expertise.

From the very beginning, EDM technology has continuously evolved. In the current landscape, EDM is no longer limited to shaping/machining alone. As part of this review, we have identified five primary and two nonstandard areas of practical electro-erosion application (Figure 3).

Wire-electrical discharge machining (WEDM) serves a dual purpose, involving both cutting and contouring processes [25,41,42]. In this method, a wire acts as the ET. WEDM is characterized by a complex iterative process that involves an unpredictable cratering mechanism [41]. The processing conditions and the physical, mechanical, and thermomechanical properties of the materials can result in wire breakage and even damage to the material itself [25,42]. Notably, WEDM offers certain advantages, such as effectively providing a constantly renewed tool in the processing zone, and relatively low tool costs [40]. However, due to the heat-affected zone, WEDM finds its primary application in roughing processes, especially in small-batch production [38].

| **Main (Standard)** | **EDM** | **Nonstandard** |

Wire-electrical discharge machining (WEDM) is a manufacturing process that is used to cut complicated shapes through electrically charged wire.

Electrical discharge machining sinking (Die-Sinking EDM) is a technology especially useful for the manufacture of stamps and molds, and with its help it is possible to accurately manufacture complex parts.

Electrical discharge machining milling (EDM Milling) aims to machine deep cavities with rotating electrodes.

Electrical discharge machining drilling (EDM Drilling) is a method widely used to create precise holes in various materials.

Micro electrical discharge machining (MEDM) is a technology that allows for the processing of micro-tools, micro-components, and parts with micro-features, due to high accuracy, low tolerance, and good surface quality.

Electrical discharge coating (EDC) is an unconventional technique wherein hard layers are obtained on the substrate surface.

Electro-erosion edge honing (EEEH) is a novel process that exploits the undesirable but inevitable phenomenon of localized electrode wear in electrical discharge machining that rapidly rounds off sharp edges for the edge honing of cutting tools.

**Figure 3.** Classification of the applications of EDM technology (the list is still evolving).

Die-sinking EDM employs electrical erosion piercing technology [7,8,25,31,40]. The concept of material removal through electrical erosion was pioneered by Boris and Natalia Lazarenko [6,7]. In the die-sinking EDM process, the tool electrodes, which come in various geometric configurations ranging from canonical to complex shapes, are used to shape and machine the surface. Electroerosive piercing is a widely adopted technique, particularly for crafting molds, stamps, and other intricate systems [43,44]. In the EDM milling method [25,45–47], a tool is rotated and moved along a specific trajectory, somewhat resembling standard milling techniques. However, EDM milling achieves the desired part shape through non-contact electrical erosion [45]. One notable advantage of this process is the standardization of electrodes [46]. The electrode's profile intricately depends upon various factors such as wear, tool path, gap distance, and material redistribution [47]. There is also an electro erosive analog of the drilling process [25,48–50]. EDM drilling is associated with the rotation of the electrode during surface treatment [48]. The main problem of this process is to create deep holes using electro-erosion [49]. In this process [25,49], low productivity of processing is noted, and poor-quality removal of material debris that arose after erosion. Surface quality and machining accuracy are also reduced due to the inaccessibility of gaps for cleaning and debris removal. The main advantage of the technology is obtaining surfaces with better accuracy and quality than with mechanical

drilling [25]. Modern research considers modification of EDM drilling technology to improve surface quality and increase the depth of material removal [49,50].

The development of micro and nano systems, driven by technological progress [51,52], has a notable impact on EDM. Specifically, micro electrical discharge machining (MEDM) plays a crucial role in creating microstructures [53,54]. MEDM technology incorporates various processing methods, including wire [55,56], sinking [54,57], drilling [48,53], and milling [47,58].

The active development of electrical erosion technologies is utilized with EDM's inherent tool wear or erosions. One such technology is electrical discharge coating (EDC), which utilizes tool wear/erosion to create functional coatings [19,26,59,60]. The quality of the coating and the percentage of the desired material depends largely on factors such as tool material and processing conditions (including dielectric flushing, electric arc, and reverse charges) [19,59,60]. Surface modification represents a new dimension of EDM [26], involving the controlled transfer of material from the tool to the workpiece as an evolving area within EDM.

Electro-erosion edge honing (EEEH) is another emerging technology aimed at altering tool geometry [61,62]. It involves rapidly rounding the sharp edges of tool electrodes to achieve the desired geometry and it can also be considered a novel aspect of EDM.

Figure 1 summarizes the technology development identified in our comprehensive review. However, the widespread adoption and advancement of EDM is primarily driven by its diverse applications, ranging from micromachines to biomedicine and spanning various industries such as aviation and shipbuilding [17,23,32,63].

In aviation, EDM is instrumental in creating components like disks, recesses for turbofan engine blades, sealing grooves, and diffusion holes [40,64–66]. The automotive industry also benefits from EDM, particularly in the production of fuel injection and cooling nozzles [50]. In microelectronics, EDM is used for creating micro-holes and micro-parts [17,67].

Gears can be cut using WEDM or created from molds made through die-sinking EDM [68,69]. Over recent decades, EDM has become an integral part of industrial production processes. Beyond precision machining, EDM has found extensive applications in biomedicine. In this context, the technology is primarily used to create unique functional surfaces with desired tribological properties [63,70,71]. These materials play a crucial role in modern implantology, with surface morphology contributing to the efficient integration of materials into natural organic structures [72,73].

Despite its prevalence, EDM is a very complex process with a large number of factors and problems that affect the final result [4]. When comparing the processing of titanium alloys, Pramanik et al. [25] noted that the formation of a remelted layer, heat-affected zones, and tool wear are typically associated with EDM process parameters. Most of the studies are dedicated to the factors influencing the process, as well as the physical-mechanical, thermomechanical, electrical, and other effects associated with the process.

*EDM Process Physics*

EDM is a thermoelectric process characterized by its complexity and unpredictability, influenced by numerous factors that can vary randomly [4]. Describing the material removal process solely based on empirical observations is impossible (Figure 4). Electro-erosion encompasses various intricate domains within applied mechanics, including thermo-mechanics [74], thermodynamics [75], crystal mechanics [76], mechanics of porous media [21,77], fracture mechanics [21], hardening mechanics [59,60], mechanics of liquids and hydrodynamics [75,78], tribology [63,70], etc. In addition to its connections with the physics and mechanics of materials and systems, EDM has ties to chemistry, electrical engineering, and materials science [4].

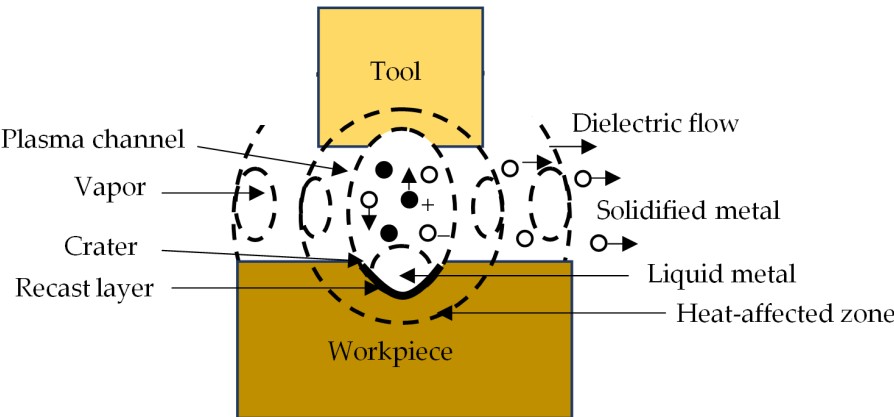

**Figure 4.** Schematic diagram of material removal in EDM.

The EDM process involves a wide range of phenomena, such as phase transitions [79–81], melt behavior [82–84], oxidation [36,50,85], residual stresses [86–90], energy transfer, absorption and dissipation [91–94], chemical reactions [75,95], microstructural changes [53,81,96], etc.

The EDM process is generally based on the melting and evaporation of material under the influence of highly concentrated electrical energy, which is converted into thermal energy. When the molten material comes into contact with the dielectric, it either becomes washed away as debris or solidifies, forming a crater. However, the removal processes of both the workpiece and the electrode depend on the specific materials and its work function. Furthermore, external factors like hardening or cryogenic treatment play a significant role [26,60,97,98].

The material removal mechanism can significantly differ from the standard electro-erosion process depending on the electrical resistivity and melting temperature of the material [28]. For instance, materials with high electrical resistance tend to generate additional heat, promoting melting, evaporation, and thermal exfoliation. Thermal spalling may also occur as a removal mechanism, especially in semiconductor materials.

When processing ceramics, aside from the typical removal mechanism, dissociation or spalling associated with material destruction can be observed [99]. The melting point, thermal conductivity, fracture toughness, and other thermomechanical properties of materials and composites can influence the material removal mechanisms in EDM. For example, ceramics, which are less resistant to tensile stresses, often exhibit characteristics like micro-cracking and spalling. In ceramic composites, material removal mechanisms can include oxidation and matrix dissolution [100].

Moreover, the parameters and factors of the technological process can impact not only the final result but also the material removal mechanism. In high-speed WEDM, for instance, anodic dissolution may be observed as a material removal mechanism [101]. In ceramic composites, depending on the strength of the discharge, it is possible to observe the breakage or fracture of reinforcing fibers and brittle fracture of the matrix [102].

## 3. Empirical Research

### 3.1. Factors Affecting the Process

The key factors used to assess the quality of the EDM process include performance parameters such as material removal rate (MRR), tool wear factor (TWR), roughness (typically measured as Average Surface Roughness or Ra), and processing accuracy [32]. The primary objective of the EDM process is to maximize MRR while minimizing TWR, all while achieving the desired surface quality. Nahak et al. offer a comprehensive review detailing how performance parameters depend on materials, processing conditions, and their mathematical interpretation [23]. These performance parameters are influenced by both electrical and non-electrical process parameters.

Electrical parameters include various factors such as discharge energy, open-circuit voltage, gap voltage, peak current, pulse duration, pulse interval, fill factor, and more [32]. Experimental studies highlight the significance of different electrical parameters, as they have a considerable impact on the final processing outcome. This is due to the stochastic nature of EDM.

In several studies, peak current has been identified as the primary parameter affecting process productivity [103–105]. Anand et al. [103] observed that peak current has the most significant impact on output characteristics across various responses. When a high peak current is combined with a long pulse duration, it results in shorter processing times and higher MRR. The effectiveness of processing with a composite electrode-tool, using materials like graphite and copper, was investigated by Ablyaz et al. [106] to enhance efficiency. Figure 5 illustrates the comparison of processing times between a copper and a composite ET.

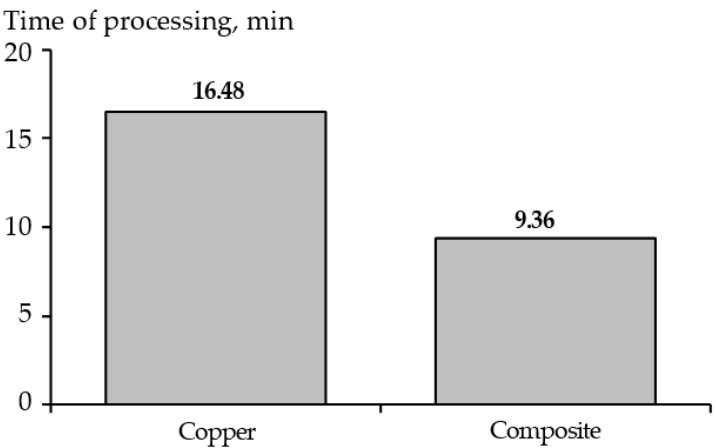

**Figure 5.** Processing time of the product with a copper and composite electrode tool.

Rizwi et al. discovered that the TWR is primarily affected by the pulse duty cycle and peak current, and these findings are confirmed by Straka et al. [104]. They reported that peak current has the most significant impact on both MRR and TWR. Peak current is a key factor influencing MRR and surface finish, whether in finishing or roughing processes [105].

Other researchers [15] have found that MRR and TWR are influenced by energy density and pulse duration, with pulse duration having the most significant effect. Singh et al. [77] obtained similar results, emphasizing that longer pulse durations increase the likelihood of carbon deposition on the electrode surface, reducing wear [107]. In their work [104], the range were established for the maximum pulse duration, beyond which EDM becomes unstable, leading to a decrease in MRR.

Many process parameters are directly influenced by the characteristics of the workpiece, electrode, and dielectric medium. For instance, the tool's geometry directly impacts the energy density of electrical discharges. Kiyak et al. [108] investigated the variations in dielectric media that affect EDM performance. Additionally, the energy density can be adjusted by increasing the diameter (interaction/active area) of the electrode tool, a trend that has been proven for decreasing electrode wear as the radius/active of the electrode tool increases.

In some cases, non-electrical parameters can have a greater impact on the process than electrical parameters. Anand et al. found that pulse duration has less influence on process performance compared to dielectric medium selection. Moreover, the temperature of the dielectric medium affects EDM process responses [109,110]. Lowering the dielectric liquid temperature to −50 °C can increase processing speed by nearly 30%, while reducing electrode wear by approximately 17%. This enhancement may be due to the brittle fracture as a dominating factor for MRR during EDM processing.

One notable research area within EDM is the machining of difficult-to-process materials, such as composite materials [111–113]. These materials often have low conductivity due to their physical and mechanical properties, like polymer composites. To process these materials efficiently, the application of a conductive layer is suggested [113]. The processing scheme for these polymers is illustrated in Figure 6.

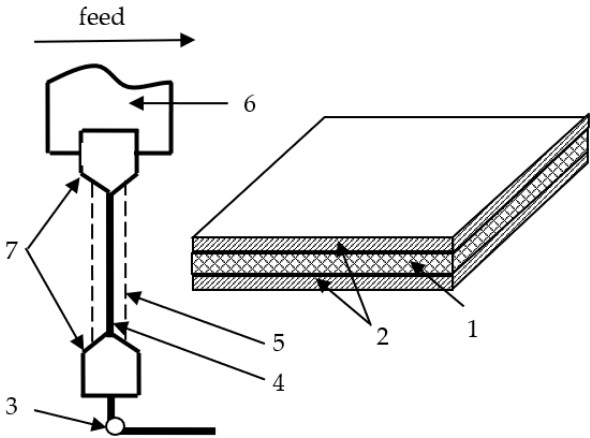

**Figure 6.** Scheme of the WEDM of polymer composite material (PCM) with conductive layers used in the present study (1: PCM, 2: conductive Ti-layers, 3: tension roller, 4: ET wire, 5: flushing, 6: gearbox, 7: upper and lower diamond wire guides) [113].

Using this technique, it is possible to obtain complex-profile surfaces of products [113] obtained from polymer composite materials (Figure 7).

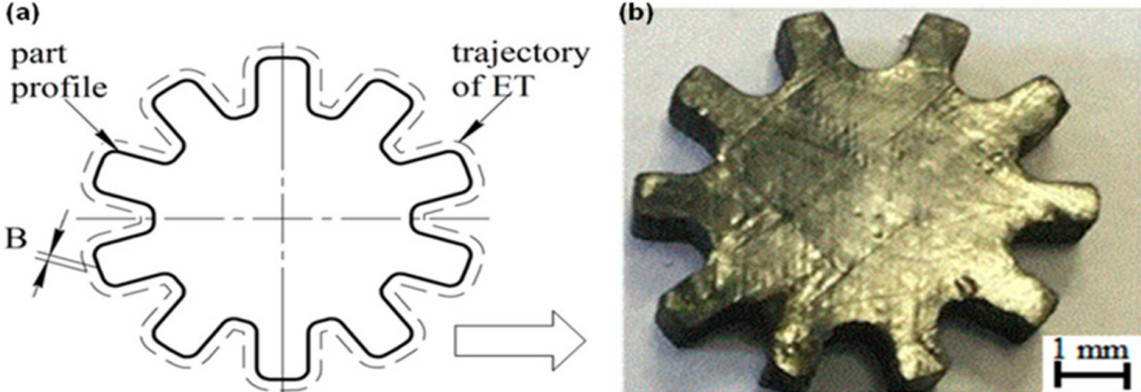

**Figure 7.** (**a**)—Trajectory of the ET, (**b**)—finished product.

Processing accuracy is a crucial factor in ensuring the quality of EDM, but it becomes challenging when machining bimetallic materials. Researchers from the Center for Additive Technologies at Perm National Research Polytechnic University (Perm, Russia) investigated the EDM processing of bimetallic materials using various ETs [114]. They found that when EDM is performed on materials with different thermo-physical properties, an uneven total energy formation within the workpiece and tool occurs. Additionally, the thermophysical properties of materials impact their electroerosive resistance. Electrical conductivity is a key factor characterizing a material's electrical erosion resistance, leading to non-uniform material removal in the form of steps. Figure 8 illustrates the formation of unevenness on a processed workpiece (bimetallic) made of steel and copper using a graphite electrode, where 'h' represents the unevenness of workpiece processing.

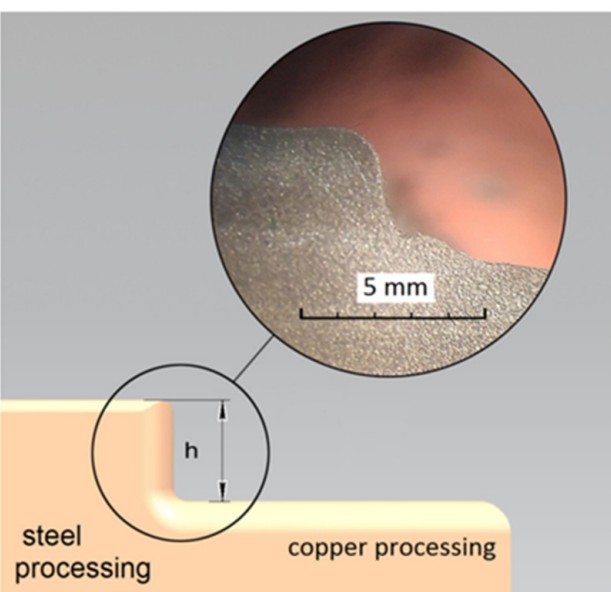

**Figure 8.** Uneven wear during EDM processing with graphite tools [114].

The performance of the EDM process is directly related to the conditions of experimentation and the material properties involved in the experiment (workpiece, tool, dielectric).

### 3.2. Experimental Data

The precision processing capabilities of EDM remain unaffected by the physical-mechanical, thermomechanical, metallurgical, and chemical properties of workpiece materials, making it suitable for a wide range of materials, including those with low electrical conductivity [115]. As materials science advances, leading to the development of new structural and functional materials with unique properties, the applications of EDM continue to expand in the industries [116]. New materials designed for extreme conditions, characterized by low thermal conductivity, high melting temperatures, and high hardness (e.g., titanium and its alloys), are challenging to machine using conventional methods. This led to the adoption of non-conventional techniques such as electro-erosion technology. EDM has become widely utilized for processing materials like titanium [4,36,117], steel [50,104,118], and various alloys [80,119,120]. Recently, EDM has also been increasingly applied to non-traditional materials, including composites [42,77,121], ceramics [29,30,107], and semiconductors [28,122]. Electrode materials must resist erosion, offer workability, and ensure stable machining. Copper and copper-based alloys, like pure copper, are commonly used for ET due to their erosion resistance and suitability for a wide range of workpiece materials [123]. As for dielectric media, kerosene and industrial oils are frequently employed [124–126]. A recent assessment of published works has provided insights into the materials used for workpieces (Figure 9a), tools (Figure 9b), and dielectrics (Figure 9c,d), along with a critical evaluation of experimental data presented in modern publications (Figure 9d).

Experimental studies of titanium and its alloys are mainly aimed at modifying the process or considering special phenomena. Hrițuc et al. considered the cutting process on titanium-coated specimens and analyzed the non-uniformity of tool wear [117]. Maddu et al. showed the formation of bronze coating on titanium using a copper electrode with preliminary hardening of the processing surface [60]. Biocompatibility [71] and tribology [63] of titanium alloys after EDM treatment are investigated. The effect of different electrode materials on MRR and TWR during the processing of a workpiece made of titanium alloy Ti-5Al-2.5Sn was analyzed [127]. Some other authors stated that different types of EDM processes influenced the machining mechanism, tool, dielectric, MRR, and surface integrity of workpieces made of titanium alloys [25]. Experimental

studies for titanium and its alloys are currently focused on the influence of the different settings of process parameters on the various characteristics of the materials and alloys.

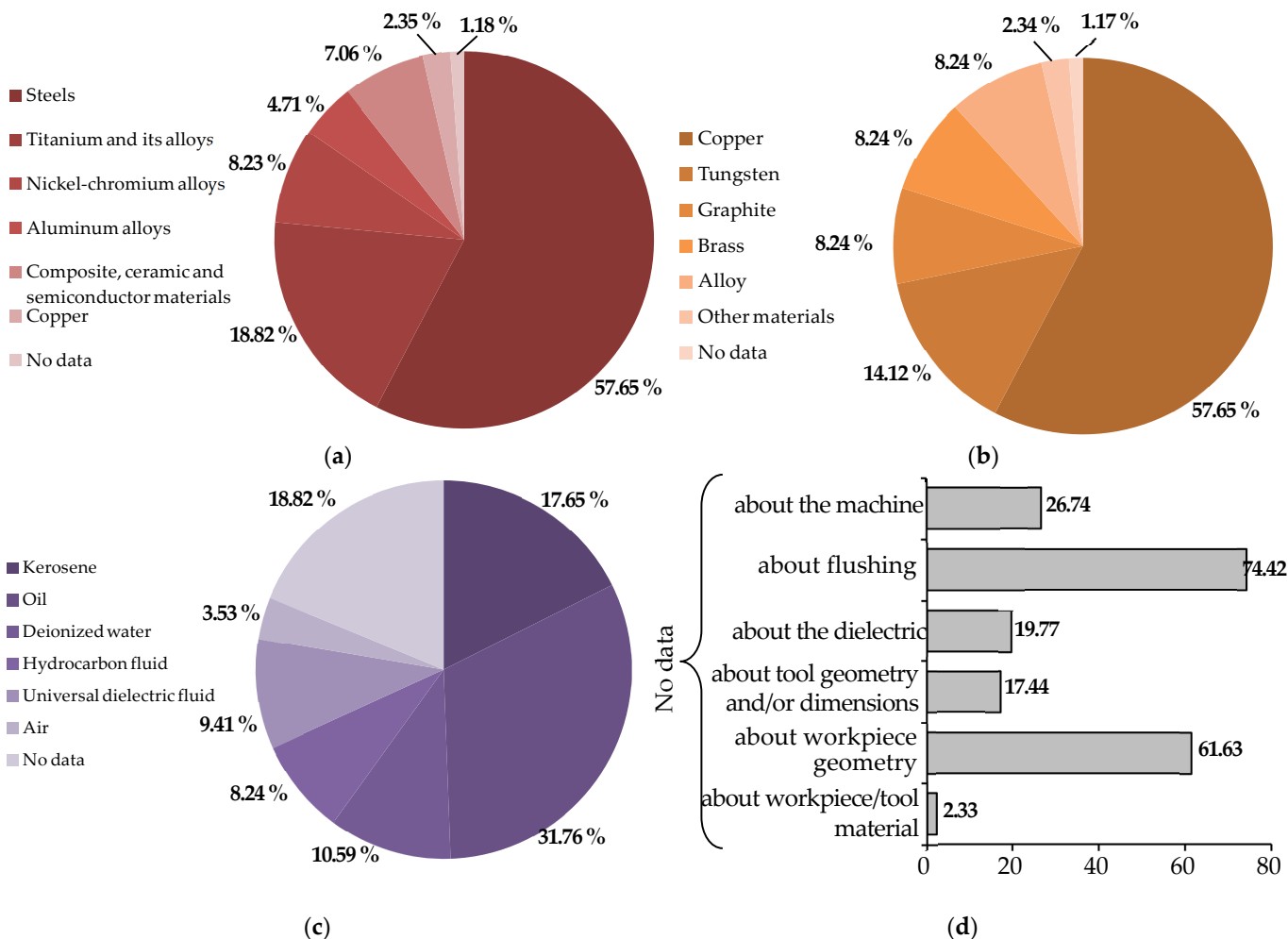

**Figure 9.** Results of evaluation of publications regarding: (**a**) is workpiece; (**b**) is tool; (**c**) is dielectric; (**d**) is missing information about experiments.

At present, the prevailing percentage of experiments is aimed at studying the processing of steel billets (Figure 9a). In general, research is aimed at selecting the optimal processing modes to obtain the required output parameters of the process [104,110,120,128,129]. The influence of the tool material and dielectric on the qualitative results of the EDM processing was also evaluated [118,120,130,131]. There are studies aimed at a comparative analysis of the surface quality in the processing of different types of steel [131]. The stages and results of the technological process of steel processing were compared with other workpiece materials, such as titanium and aluminum alloy [71]. Crack formation/surface integrity was also studied by the researcher [132]. A fairly large percentage of research is devoted to the analysis of EDM processing of workpieces made of nickel-chromium [98,133,134] and aluminum alloys [80,120,124].

There are also non-traditional experimental studies involving copper on copper [135] and tungsten on copper [136]. Also, workpieces are subjected to various modifications through heat treatment [30], hardening [26,60], etc. Surface modification and processing of materials suitable for standard methods is necessary for the required application of materials.

More than 57% of the studies considered in the review were performed using copper electrodes (Figure 9b). In the second place, there are electrodes made of tungsten or alloys based on it (about 14%). Graphite, brass, or aluminum alloy electrodes are used in approximately 8% of the experiments. Studying the influence of the tool material on the nature of workpiece surface treatment is still relevant [70,77,120,130]. The influence of the tool polarity on the EDM is also examined [130]. However, the prevailing number of studies related to ET is aimed at reducing their wear or compensating for modification. It is possible to single out the main directions of modification: selection of new non-traditional materials [17,103,137,138], processing or coating [17,97,98,139–141], creation of non-traditional geometric shapes [49,137,142–144], using modern methods of additive manufacturing [145–148], creation of original structures [146,148], etc.

To compensate for wear, diamond [17], composite [139], and highly heat-resistant steel [140] coatings or cryogenic treatment is used [17,97,98]. Improvement in flushing and processing accuracy was achieved by creating a stepped geometry of the electrode-tool, but this led to a high roughness [49,50]. A change in the sparking surface is observed, associated with the non-standard geometry of the electrodes (providing relief angles and a cylindrical surface of different thicknesses at the end of the electrode) [137]. This leads to a slight improvement in the results of the technological process. Though, by varying the geometry, it is possible to achieve a decrease in the taper of the holes up to 10%. Reducing tool wear is also achieved by applying tungsten carbide (WC) coating–thin films with better electrical and thermal properties [141]. The use of additive technologies to create tools does not always give effective results. For example, an electrode obtained by additive sintering has a lower MRR with a greater degree of tool wear and a higher level of roughness than a standard tool fabricated using the conventional routes [145]. On the contrary, a selectively designed porous electrode allows for recovery after use and a significant reduction in wear due to the improved flushing quality of the corrosion products [146]. This type of electrode can be used quite effectively for roughing (machining time is 47% lower than that of a standard electrode).

Recently, studies have been carried out that are related to the rational, from the point of view of ecology and productivity, selection of dielectric liquid [98,124,134]. The effectiveness of the work and the possibility of introducing various biodegradable liquids and oils into the process are being actively evaluated [21,27,36,124]. It is found that when using kerosene, a protective carbide layer is formed on the electrode surface, and when using distilled water, on the contrary, an oxide layer is formed, which easily comes off and contributes to tool wear [124]. When using distilled water, the quality of the treated surface reduces due to a large number of cracks and defects. In general, due to the increased MRR, distilled water has better dielectric properties and is safer for the environment as compared to kerosene. Modification of the dielectric medium through various additives is also aimed at improving the EDM process [26,97,126,131].

In a significant number of works reflecting experimental studies of the process, there is a lack of data: data on flushing (>74%), geometric parameters of the workpiece (>61%), machine information (>26%), etc. At the same time, almost all publications indicate the materials of the workpiece/tool. Only less than 3% of publications address the development or modification of the EDM process.

Figure 10 shows general statistics for experimental studies. The main parameters varied in the studies and the output parameters that were monitored after the EDM are considered.

Global areas of research are workpiece wear (73%) and tool wear (52%). The percentage of studies on the nature of tool wear, creating methods for predicting and compensating it, including those using mathematical and numerical models, is increasing. In total, 25.3% of publications examine both workpiece wear and tool wear. This fact is associated with the need to predict workpiece wear, taking into account the influence of tool wear on it. The prevailing number of experiments is still associated with varying the parameters of the

technological process parameters ~79%. The main parameters to observe during and after the experiments are MRR, EWR, TWR, Ra, and surface morphology.

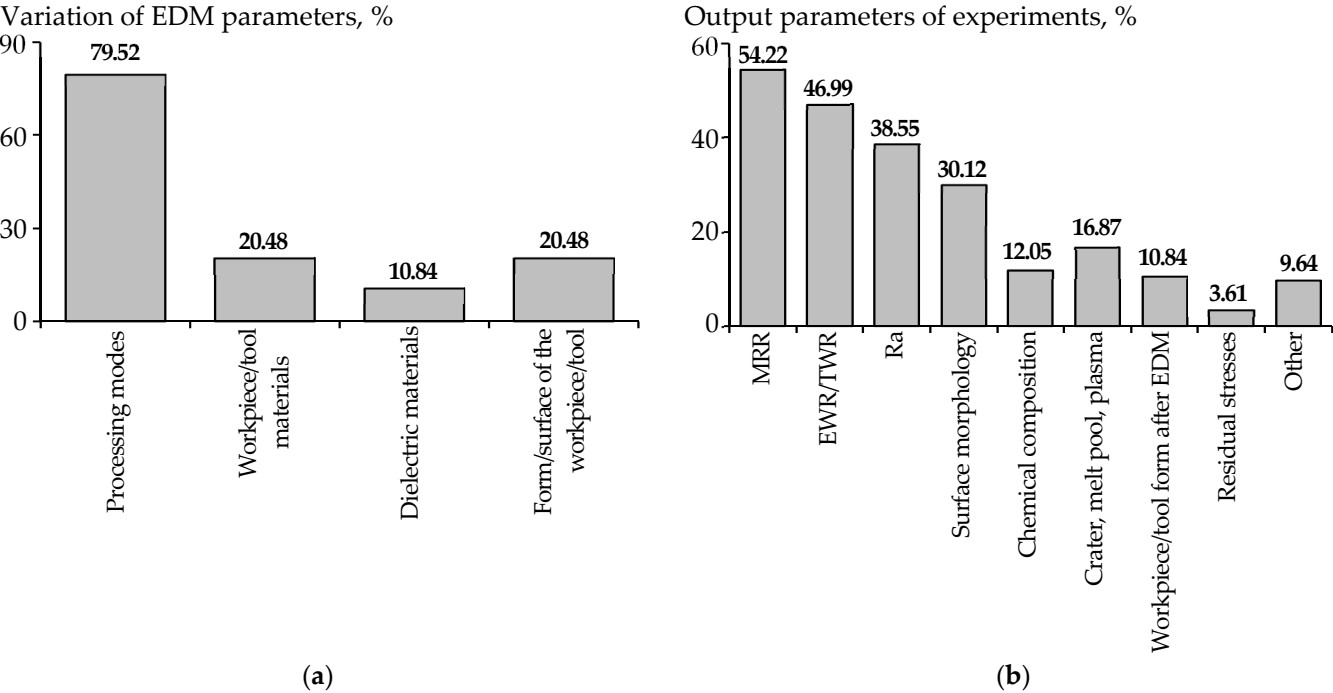

**Figure 10.** Quantifying experimental parameters in research: (**a**) main parameters of variation; (**b**) output parameters.

The main problems of the experimental base can be noted as rapid tool wear, debris and by-products of erosion, cracks, and other defects. To be able to eliminate and compensate for them, statistics and optimization of the experimental base, as well as effective mathematical and numerical models, are required.

## 4. EDM for Surface Engineering of Biomaterials

Cobalt-chromium, titanium, magnesium, and stainless-steel alloys are examples of metallic materials that are often chosen as candidate materials for a range of biomedical applications, such as orthopedic implants that must withstand loads and cardiovascular medical equipment. Their exceptional biomechanical, corrosion-resistant, and biocompatibility qualities, which qualify them for application in the production of biomedical devices, are the primary causes of this feature. But since these biomaterials are biomedical devices, their surfaces play a critical role in the ongoing interactions between the surface of an implanted metallic substance and the surrounding physiological environment. Therefore, in order to guarantee these metallic material's dependability and effectiveness as biomedical devices in the body's physiological environment, further surface treatments are essential and EDM is emerging as a potential candidate for surface alteration. Without altering the metallic material's bulk characteristics, surface changes on these biomaterials with metallic qualities control the biological response [149,150]. The attenuated surfaces of these biomaterials are known to emit hazardous chemicals including V, Cr, Ni, Co, Al, and Fe, despite their excellent mechanical strength and resilience to wear and corrosion [151]. The solution to these adverse effects can be alleviated with applications of powder mixed EDM. The element transfer mechanism during PM-EDM is represented in Figure 11 and Table 1 lists the commonly used conductive additives in PM-EDM for biomaterials fabrication and surface modification for enhanced biocompatibility.

Electro-discharge treatment (also known as electro-discharge coating or EDC) is among the most recent and efficient route to alter the bio-metallic surface characteristics by mixing the bioactive powder (known as PM-EDM) in dielectric amid other developed processes such as sol-gel, electroplating, laser coating, PVD, CVD. In order to modify the surface using EDT, the fundamental principle is to choose an ideal combination of process parameters that alters the spark energy produced within the electrodes. In Pm-EDM, the most significant control variables influencing the bio-favorable surface were dielectric, pulse duration, and current for the surface treated [152].

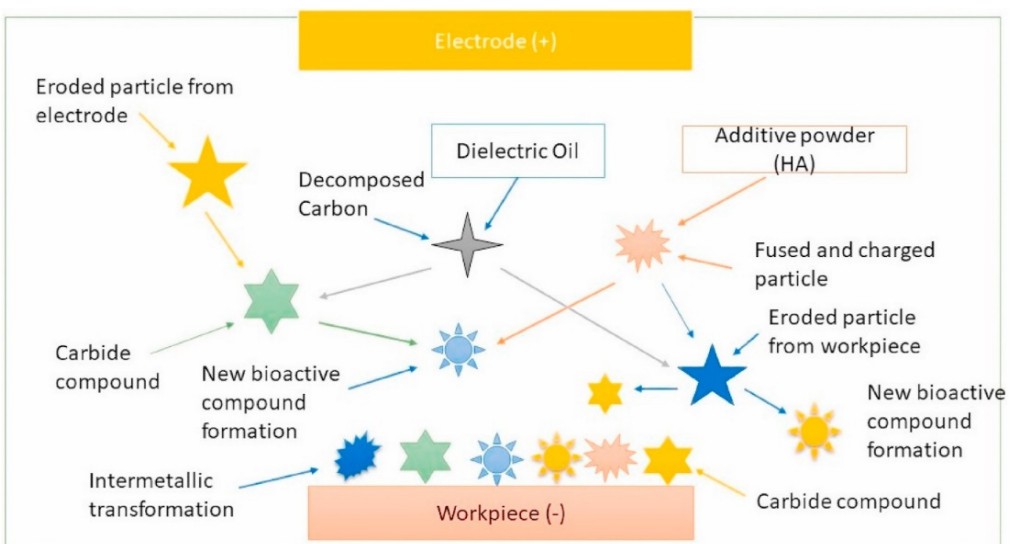

**Figure 11.** Mechanism of material migration and alloying during PM-EDM process. Reproduced with permission from Elsevier [153].

**Table 1.** Commonly used conductive additives in PM-EDM cycle for biomaterials fabrication. Reproduced with permission from Elsevier [153].

| Additives | Density (g/cm$^3$) | Fusing Point (°C) | Electrical Resistivity (μ-ohm-cm) | Thermal Conductivity (W/mK) | Molecular Weight (gm/mol) |
|---|---|---|---|---|---|
| Hydroxyapatite (HA) | 3.08 | 1250 | Higher | 0.01 | 502.31 |
| Silicon | 2.33 | 1410 | $6.00 \times 10^7$ | 150 | 28.08 |
| SiC | 3.21 | 2987 | $1.00 \times 10^9$ | 500 | 40.11 |
| Aluminum | 2.7 | 660 | 2.45 | 238 | 26.982 |
| Titanium | 4.72 | 1668 | 55 | 22 | 47.88 |
| TiO$_2$ | 4.23 | 1855 | $1.00 \times 10^{24}$ | 7.4 | 79.866 |
| CNT | 2 | 2800 | 50 | 4000 | N/M |
| Graphite | 1.95 | 4550 | 500–3000 | 470 | 12.011 |
| Silver | 10.49 | 962 | 1.6 | 430 | 107.86 |
| Zinc | 7.13 | 420 | 5.5 | 120 | 65.38 |
| Copper | 8.96 | 1083 | 1.59 | 416 | 63.54 |
| Chromium | 7.16 | 1875 | 2.6 | 67 | 51.996 |
| Molybdenum | 10.2 | 2610 | 5.27 | 139 | 95.94 |
| Silicon | 2.33 | 1410 | $6.00 \times 10^7$ | 150 | 28.08 |

Using nano-HAp in the PM-EDM cycle, Prakash et al. [154] examined the properties of the modified surface of the magnesium alloy. According to their research, adding HAp during machining increased the treated surface's microhardness and corrosion resistance by 1.5 times and 90.85%, respectively. Moreover, the development of intermetallic oxides was shown to enhance the machined surface's biocompatibility. According to S. Santosh et al. [155], adding graphite powder shortened the breakdown voltage and increased the electrical conductivity of EDM oil, which improved MRR. It was found that ZM21 magnesium alloy increased biocompatibility. The machined surface qualities of 316L stainless steel and the machining capabilities of PM-EDM with additional hydroxyapatite powder (HAp) were examined by Yubraj et al. [156]. In the case of the machined 316L stainless steel with copper electrode, the addition of additive powder, current, and pulse-on time had the greatest effects on surface roughness. The formation of oxides and intermetallic compounds, such as calcium-phosphate-zinc, iron-molybdenum, zincite, calcium carbonate ($CaCO_3$), iron-silicate-carbide, phosphorus (P), and calcium (Ca), confirmed the presence of a biocompatible layer; however, the impact of powder concentration and other related factors were not examined. With a continuous reverse polarity of the copper electrode, Singh et al. [152] examined the impact of the HAp mixed PM-EDM process on the surface characteristics of machined 316L stainless steel. An increase in MRR of 19.01 g/min was seen in this investigation at a current of 28A, a pulse-on time of 120 μs, and a powder quantity of 15 g/L. The most influential factors for surface coating were found to be the current, followed by the pulse-off and pulse-on durations. Using the PM-EDM approach, Singh et al. [157] altered the surface of 316L stainless steel by adding $TiO_2$ powder and tungsten (W) electrode. Due to the creation of carbides, silicides, and intermetallic alloys, the addition of $TiO_2$ particles and subsequent current had a huge impact on improving the treated surface's microhardness and wear resistance. The wear rate was reduced by 80% at a current of 28A with additional $TiO_2$ particles, exhibiting an ideal hardness of 942.90 HV with an increase of 223% compared to the untreated sample. However, the quantity of added powder was not made clear. The use of a greater discharge current resulted in microcracks even with the inclusion of $TiO_2$ powder to the process. Moreover, XRD analysis was used to assess the existence of bioactive components such as Ca, P, and $CaCO_3$.

Using HAp in the PM-EDM process, Bains et al. [158] assessed the MRR and coarseness of the treated surface of Ti–6Al–4V. It was shown that pulse-on and pulse-off length were relatively less significant determinants for MRR, while both current and the addition of HAp were significant factors. While the addition of nano-HAp to a current of 12 A increased MRR by about two times, the response induced by rising surface hardness owing to carbide and oxide formation was partially mitigated when the current amount was shortened by 6 A. The effect of PM-EDM process parameters on Ti6Al4V alloy machining using suspended silicon powder was investigated by Nipun and Anand [159]. The three strongest arguments in favor of MRR were concentration of Si powder, pulse-on time, and current. With increases in current, pulse-on time, and Si powder concentration, MRR was shown to spike; however, beyond a certain point, MRR decreased as a result of inadequate debris flushing. The impact of carbon nanotube (CNT) powder-mixed EDM on MRR and TWR for Ti-6Al-4V alloy was investigated by Mohammadreza et al. [126]. While prior research has shown that adding CNT to the dielectric liquid may increase the machining time by 66% when it comes to steel alloy machining, this study found that adding CNT to the dielectric liquid reduced both MRR and TWR. The process parameters for machining Ti-based alloys using reverse-polarity graphite electrodes and MWCNT suspended in a dielectric liquid were optimized by Bhui et al. [160]. MRR increased by 37% in this investigation when the WMCNT suspended dielectric was used. A rise in MRR was seen at 4 A current, 60 μs pulse-on time, and MWCNT addition; however, the quantity of CNT powder added was not specified. With an increase in current, TWR rose. In the PM-EDM technique, Devgan and Sidhu [161] examined the treated surface of β-type titanium by adding MWCNT to HAp-mixed dielectric medium. While the

surfaces treated with HAp and MWCNT demonstrated twice as much cell survival as the untreated surface, adding MWCNT to the HAp-mixed dielectric was preferable for improving biocompatibility. Using nano-aluminum particles in a dielectric liquid during an EDM cycle, A.M. Abdul-Rani et al. [162] investigated the surface morphology and roughness of the machined titanium alloy. Comparing this investigation to the conventional EDM technique, a little improvement in surface morphology—that is, in terms of cracks and voids—was seen. In comparison to the traditional EDM, the addition of 3 g/L of nano aluminum particles lowered SR by about 38.46%. Higher SR was caused by non-uniform distribution and agglomerated additive particles.

A unique method for changing the titanium alloy's surface using silicon additions was used in the work by Chander et al. [163]. In this work, a 15 μm thick recast layer consisting of carbides and oxides at 15 A current, 50 μs pulse-on time, and 8 g/L Si powder quantity was formed, resulting in the fabrication of a biocompatible and hard surface of 1080 HV. Furthermore, the treated surface's wear rate increased by about 29 times and its coefficient of friction decreased by 60% as a result of the TiC, NbC, and SiC production. By combining Si powder in the PM-EDM process, Farooq et al. [164] examined the surface morphology, SR, and RLT of the titanium alloy. By 61.82% and 37.03%, respectively, the Si additive concentration helped to regulate SR and RLT. Applying 5 g/L Si powder reduced the SR by 3–3.5 μm as compared to 0 g/L; however, increasing the quantity of powder from 10 g/L to 20 g/L resulted in an increase in SR above the 5 g/L concentration. Using HAp as a dielectric, Chander and Uddin [165] investigated the surface properties of the modified Ti alloy. At 5 g/L HAp concentration, a modified surface with microcracks, voids, and ridges was detected, but at 10 g/L HAp concentration, the surface was smooth and devoid of fractures. Elevating the concentration of HAp led to a three-fold improvement in microhardness, corrosion resistance, and biocompatibility due to the faster deposition of oxides and carbides on the machined surface.

It is clear that there are still certain problems with the PM-EDM cycle that need to be addressed before bio-implants are produced in large quantities for the industries. More thorough research should be conducted on these areas. Figure 12 shows the future research directions for the PM-EDM process.

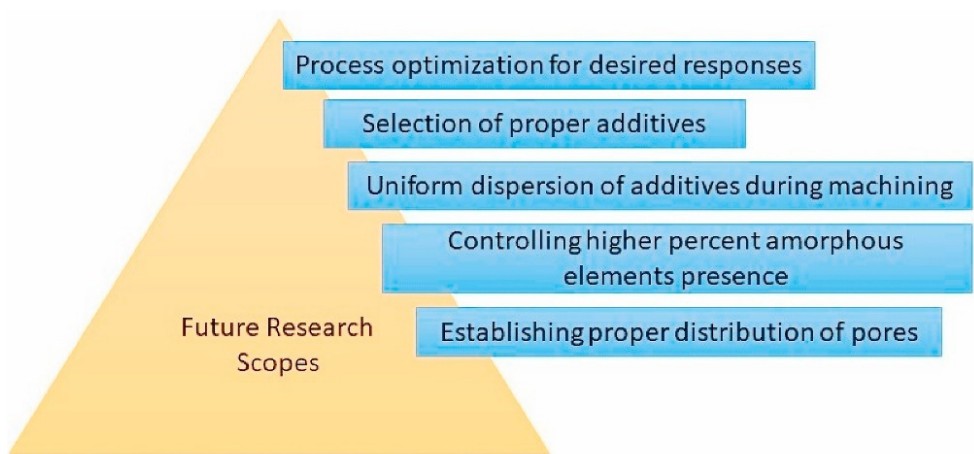

**Figure 12.** Future research directions of PM-EDM. Reproduced with permission from Elsevier [152].

Selected patents were thoroughly reviewed and presented in Table 2. Table 2 collects the patents related to the EDM process and shows the information under the headings: (1) Publication No., (2) Title, (3) Inventors, (4) Assignee, (5) Publication Year.

**Table 2.** Table related to the EDM process.

| Publication Number | Title | Inventors | Assignee | Publication Year | Ref. |
|---|---|---|---|---|---|
| RU2802609 | Device for electro discharge firmware opening with electrode tool | Ablyaz T., Shlykov E., Osinnikov I., Makarova L., Muratov K., Shiryaev V. | Perm National Research Polytechnic University | 2023 | https://www.fips.ru/iiss/document.xhtml?index=2 (accessed on 15 November 2023) |
| USOO9452483B2 | Electric discharge machining die sinking device and related method of operation | Yuefeng Luo William Edward Adis Michael Lewis Jones | General Electric Company, Schenectady, NY, USA | 2016 | https://patents.google.com/patent/US9452483B2/en?q=(EDM)&oq=EDM (accessed on 15 November 2023) |
| RU2772410 | Method for electric discharge wire cutting machining | Ablyaz T., Shlykov E., Gashev E., Muratov K., Shiryaev V., Sarabdjeet S. | Perm National Research Polytechnic University | 2022 | https://www.fips.ru/iiss/document.xhtml?faces-redirect=true&id=c8eacdeff9e50bcb556e0f9ed05bfc04 (accessed on 15 November 2023) |
| RU2730321 | Method of electro-discharge hole sewing | Ablyaz T., Muratov K., Makarova L., Shlykov E., Shipunov G., Shakirzyanov T. | Perm National Research Polytechnic University | 2020 | https://www.fips.ru/iiss/document.xhtml?faces-redirect=true&id=d3675ea60d25613f4a768a3c388ea401 (accessed on 15 November 2023) |
| RU2721245 | Method for texturing a metal surface | Ablyaz T., Muratov K., Makarova L., Shlykov E., Kochergin E. | Perm National Research Polytechnic University | 2020 | https://www.fips.ru/iiss/document.xhtml?faces-redirect=true&id=02aea5400721fe9315457f9c9b8829ae (accessed on 15 November 2023) |
| ES2750567T3 | Small hole EDM machining apparatus | Suzuki E., Shimoda Y. | Astec Co., Ltd. (Tokyo Japan) | 2020 | https://patents.google.com/patent/ES2750567T3/en?q=(EDM)&after=publication:20150101 (accessed on 15 November 2023) |
| US8629368B2 | High-speed ultra precision manufacturing station that combines direct metal deposition and EDM | Jyoti Mazumder Jun Ni Albert Shih | DM3D Technology, LLC, Troy, MI, USA | 2014 | https://patents.google.com/patent/US8629368B2/en?q=(EDM)&oq=EDM (accessed on 15 November 2023) |

## 5. Optimization of Process Parameters

To obtain the required surface quality of the workpiece with minimum defects involves optimizing the process parameters [92,103,105,166–168]. This problem reduces the selection of the optimal input parameters for a specific material [92,103,105,166,168] and/or the processing method and/or the nature of the process [166,167]. For optimization, intelligent process models based on neural network technologies [169,170], fuzzy logic methods [168], experiment planning methods (Taguchi method) [166], analysis of variance (ANOVA) [167], surface response methodology [167], and complex computational-computer-based methods [122] are applied. It is noted that there are few studies devoted to comparative analysis of optimization efficiency [170] which leads to solutions close to optimal or suboptimal. However, there are studies aimed at evaluating methods for optimizing the input parameters for the final result. Studies have shown that teaching-learning-based optimization (TLBO) obtains nearly global optimal solutions for single- and multi-objective optimization problems. TLBO is the least affected by both its own settings and process parameters. A combination of methods or a hybrid optimization approach is also being used [167].

The most common optimization of process parameters is using the Taguchi method [124,128,140,141,167,171,172]. It is based on the theory of experimental design and elimination of manufacturing defects before they occur, and it is also based on the quality loss function [173]. The uniqueness of the Taguchi method lies in its efficiency [174] for achieving adequate optimization results with a relatively minimal number of experimental runs. When optimizing a technological process or experiment, the approach of design matrix is used [175]. The parameters that affect the process and their levels are determined. As part of changing the levels of parameters, an orthogonal array of experimental data is built. An optimization procedure is based on an orthogonal array. It determined which controlled variables have the greatest noise and impact on the objective function and calculated as signal-to-noise ratio. The Taguchi optimization method is applicable in uncontrolled or difficult to control conditions [176]. Thus, it is well suited for the stochastic EDM process. The Taguchi method is often used for ANOVA analysis of variance [124,167,171] and Gray relational analysis [141,177].

Analysis of variance (ANOVA is used as a statistical method for assessing the significance of process parameters or experimental parameters [110,124,168,171] and for optimizing process parameters [175,178,179]. As part of the optimization, the ANOVA method groups the experimental data according to the objective function and evaluates the variance of the results and the relationship between the parameters and process responses. ANOVA supports three optimization methods: analysis of parameter variation relative to total process variation [179], Pareto analysis [180], and Sobol sensitivity analysis [175]. ANOVA can be used both as part of regression analysis and in combination with surface response methodology [167,171].

It is noted that modern research is moving in the direction of advanced optimization techniques and often uses a hybrid combination of several methods for planning and optimizing an experiment [16]. The combination of Taguchi theory and ANOVA analysis of variance is most often found in EDM.

Within the framework of EDM optimization, problems are considered at different levels: single-factor [140], multi-factor or multi-parameter [24,124,171], single-objective [24], and multi-objective [124] analysis.

## 6. Mathematical Description of the EDM Process and Numerical Simulation

The enhancement of the EDM process efficiency is considered not only in terms of experiment optimization through planning and selecting optimal processing parameters. From the 1970s to the present, developments have aimed at mathematically describing the process and providing analytical solutions to its actual problems [11–13,82,181–184]. The initial models are based on thermo-mechanics and thermal physics of the process, specifically addressing thermal or electro-thermal problems with a disk heat source [181,182]. Over time, the analytical description and mathematical modeling of the EDM process

began to consider erosion from the perspectives of the cathode, anode, and plasma with a point heat source [11–13]. For a considerable period, modeling a single crater with a canonical heat distribution was the focus. According to Shahri et al. [82], the Gaussian heat distribution is the closest match to the EDM process. Some analytical models consider the Gaussian distribution of heat from an electric spark [185]. Modern analytical and semi-analytical methods not only describe the wear of the workpiece but also the wear of the tool for one or many craters [183,184]. The primary direction of analytical methods involves mathematically describing individual phenomena of the EDM process depending on the parameters of the technological process [48,119,122,183,184,186]. Regression analysis is most often used for this purpose [117,167,183]. Currently, phenomenological models are also described, encompassing changes in the structure of the material and phase transitions during EDM [79]. Nevertheless, modeling of the EDM process is most often implemented using modern numerical methods and applied engineering analysis systems [133].

*Numerical Simulation of the EDM Process*

The processing depends on a large number of factors of different mechanical nature, which leads to the complexity of modeling the technological process of electro-erosion [18]. There is also poor repeatability and low manufacturability of experiments associated with the stochastic nature [187]. To improve the quality of processing, efficient numerical models and methods are required that will allow us to consider the process not only in statics but also in dynamics [84,97,187].

Initially, the main developments related to the numerical simulation of the process of electrical discharge machining of materials were based on the geometric modeling method (GMM), the finite element method (FEM), and the finite difference method (FDM) [10,82,115,133,188–191]. GMM is associated with the construction of workpiece and tool surfaces within the framework of micro-geometry (shape deviation, roughness, etc.) and macro-geometry (object dimensions, shape, etc.) [10,189]. The method is based solely on changing the geometry [187] wherein metal removal and tool wear consist of the successive removal of surface layers on the tool and workpiece. FEM and MCR refer to mesh methods of analysis or approximation methods [192–194]. The finite difference method is based on the approximation of the derivatives included in the differential formulation of the problem, and the numerical implementation of the solution search algorithm in an explicit or implicit form [193]. The finite element method is the discretization of the object under study into a finite set of elements and a piecewise-element approximation of the studied solution functions that depend on spatial coordinates and time [194]. In the numerical simulation of the problem using the FEM and MCS, the distribution of heat is considered within the framework of thermal approaches [76,79,188,195]. Such models have several disadvantages [76,188,189]: the exclusion of electrodes from the model or the invariance of their geometry, constant meshes of the processed surface that do not change in time, canonical geometry, the simplest heat distribution mechanisms, absence of material behavior dynamics (melting and solidification process), etc.

Over the past decades, process modeling has been developed using FEM, molecular dynamics model (MDM), and field model (FM) [18]. Moreover, FEM models have recently been widely developed [133,191], including those using the ANSYS (ANSYS Inc., Canonsburg, PA, USA) [110,196], ABAQUS (Simulia Corp., Providence, RI, USA) application software packages [81], and COMSOL Multiphysics (COMSOL Inc., Burlington, VT, USA) [197]. However, the MCE cannot investigate the formation and evolution of defect structures during a discharge at the atomic level [76]. This fact contributed to the involvement of MDM in the realization of EDM problems. The methods are based on numerical simulation of the trajectory and interaction of atoms and molecules [18,76,83]. Simulation of an electric field or phase transitions of a material structure is carried out using the phase field method (PFM) [18,79,198]. That expands the possibilities of numerical implementation of the physics and electro-mechanics of the process. There are also developments related to the involvement of artificial intelligence technologies and methods in

EDM modeling [20,54,169]. Due to the complexity of the integral-differential mathematical formulation and mathematical physics of the EDM process, problem modeling in the framework of the boundary element method (BEM) is only gaining momentum [144,199]. The considered numerical methods can be used in EDM modeling, but have their areas of application, and their advantages and disadvantages. A combination of different methods is also considered in the numerical simulation of the process [54,79,169].

As part of the work, the application of numerical modeling methods to describe the EDM process and the main achievements in describing the phenomena and environments of the technological process are assessed (Figure 13).

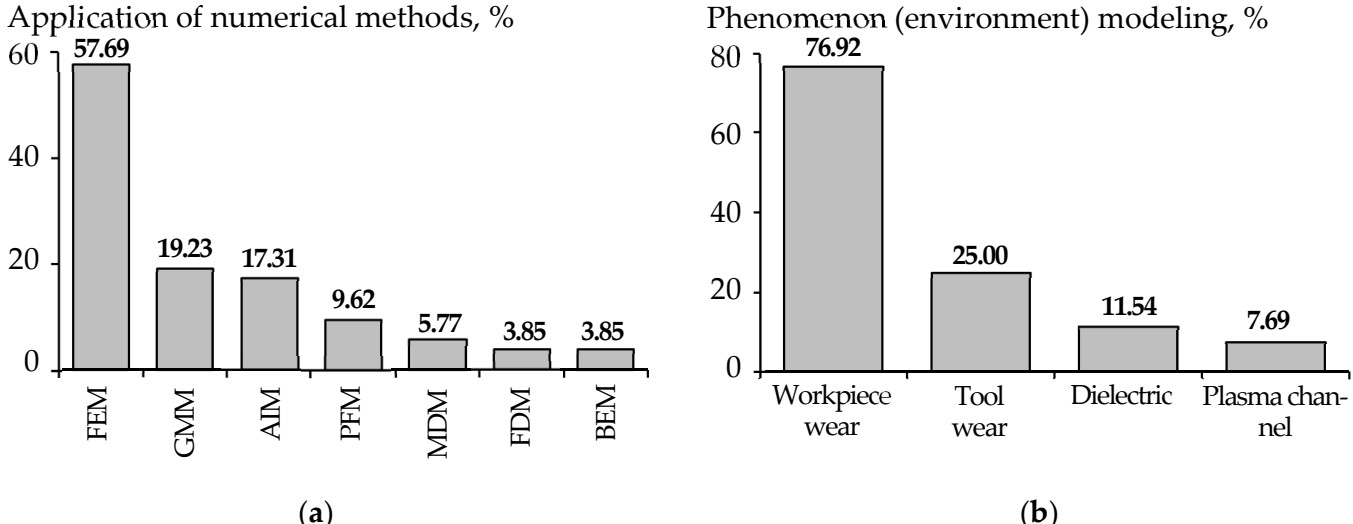

**Figure 13.** Numerical modeling of EDM: (**a**) numerical methods; (**b**) modeling the influence of various phenomena and environments on the process.

The physical phenomena in the EDM process are quite complex. For high-quality modeling, it is necessary to take into account the physics of the entire process, which at the moment is not possible to do with existing models and numerical algorithms [18]. The generalized model of the process should include [189]: discharge localization, plasma simulation, temperature distribution and material removal simulation from a single discharge and a series of discharges, gap flow simulation, object geometry simulation and its change, facility control, etc. It is required to create efficient algorithms for predicting tool wear and its compensation and monitoring [200]. The need to study the evolution of the melt pool for morphology, roughness, and surface quality during EDM is noted [84]. Processing efficiency can be increased through a fundamental understanding of the interaction between plasma and workpiece and tool materials [197]. Scientific developments in all directions are actively being carried out at present.

One can note the development of numerical simulation of electro-erosive machining associated with describing a number of phenomena [189]. Figure 11 shows the development of numerical models based on the literature data reviewed. Most modern works are taking into account the wear of the workpiece within the framework of the technological process [10,41,54,58,62,76,79,83,84,86,91,110,115,135,187,188,191,197,201–205]. The main technology used in modeling surface changes is the "killing" of elements when a certain material melt temperature is reached with the formation of a new surface type [54,133,135,191,201]. There are studies related to modeling the process of removing material by obtaining the actual energy of the discharge during a spark [93]. Ni et al. [204] reported the research in which the material was removed according to the algorithm associated with the minimum gap leading to the discharge. Such works are most often based on the FEM. But there are also improved models based on the FDM, which take into account the instantaneous material removal during the electric discharge [86]. GMM models assume a change in

the boundaries of bodies [61] wherein all mesh nodes move simultaneously in the direction normal to the surfaces of the electrode-tool and workpiece at a distance determined by the tool and workpiece wear coefficient. Research aimed at modeling the wear of a workpiece directly intersects with modeling of shaping, residual stresses, and phase transitions [79–81,86,89].

The issues of electrode wear have been dealt with for a long time. At the moment, numerical models are actively developed that take into account tool wear during machining [10,58,61,82,115,187,198,203,204,206–208]. Tool wear is the main disadvantage of the process, which degrades the topography of the workpiece surface [200]. But so far it has not been possible to solve all the problems related to the possibility of predicting the final result of electro-erosive machining, taking into account the electrode wear and its compensation [187].

Some models considered the influence of the behavior of the dielectric on the technological process [133,144,146,199,207,209]. The main areas of research are aimed at modeling the flow of a dielectric medium [133,146,207,209] and related phenomena associated with the hydrodynamics of the process [144,199]. The process of evolution of the molten material under the influence of the flow field with the formation of hardening and the crater geometry is also considered [209].

There are attempts to simulate plasma and plasma channels [86,190,205,210]. However, the plasma flow is often modeled in terms of pressure or heat [86,205].

It can be noted that a fairly large number of studies consider the formation of a single crater from a single discharge [41,62,76,79,82,133,135,184,185,197,202,210–212]. Construction of more complex models of shaping is impossible without understanding mathematical description and modeling of the process of crater formation as a whole [202]. There are digital solutions associated with the modeling of multi-pulse processing [62,203,213]. But modeling of multi-pulse processing and surface roughness is carried out by superimposing single craters on top of each other [202].

The digitalization of EDM is also aimed at creating simulation models of the process and virtual simulators [48,79,83,84,135,203]. Thanks to simulation models, it is possible to predict the geometry of the workpiece or tool profile and determine in advance the process parameters for regulation and control [48,135]. Simulation models are used to analyze the influence of individual EDM phenomena on the process as a whole [79,84]. The main direction of using such models is selection of technological process parameters to obtain a high-quality processing result with the required surface morphology parameters [83,135,203]. To select the parameters of the technological process, numerical methods for solving inverse problems are also used [94,115,188,204].

Despite the extensive data on the processing of different materials with different electrodes, there is no full-scale comparison of the quality indicators of processes with new electrode materials or electrodes subjected to modification [97]. Standardization and optimization, as well as the introduction of new technologies in the processing, have not been sufficiently developed and studied. In many ways, the widespread introduction of new technologies and materials in EDM is hampered by the lack of numerical or simulation models that closely enough reflect the entire physics of the process. Most of the existing models have rather large simplifications and assumptions [48,58,82,135,187,198,205,210]. Further development of numerical and mathematical analogs of the process is necessary to increase its efficiency.

## 7. Process Modification and Development Directions

The current state of EDM and its promising development directions are often focusing on hybrid technologies [19,31,53,115]. These technologies introduce additional influences and phenomena to enhance workpiece material removal quality and rate while reducing tool wear. In accordance with study specifics, two primary EDM development directions are identified within the context of industry 4.0 and 5.0, and these are smart manufacturing [116] and green manufacturing [31].

### 7.1. Green Manufacturing

The intensification of competition and the accelerated pace of production using EDM have heightened environmental concerns. Examining the process through the lens of sustainable development with a triple outcome—economic, ecological, and societal—has gained relevance [9,214]. Economic concerns encompass energy consumption, electrode preparation, and dielectric fluid management. Environmental issues involve hazardous emissions and the disposal of dielectric fluid. Social factors include health risks such as respiratory and skin diseases among workers associated with EDM, as well as the potential for fire or explosions, posing a significant threat to operators and other living beings [21].

With the advent of industry 5.0, technological processes are now scrutinized not only for efficiency but also for safety. Rationalizing these processes aims to reduce emissions and energy consumption while transitioning to environmentally friendly processing methods [215]. Green EDM production, which relies on biodegradable dielectrics, presents a promising avenue for the industry's future [27]. The prevalent use of kerosene as a dielectric poses significant risks to human health and the environment. To address these concerns, researchers are exploring the enhancement of processing efficiency by introducing powders into the liquid while replacing kerosene with safer dielectric alternatives. Additionally, the adoption of various oils as dielectric mediums to improve efficiency and reduce harmful emissions is a current focus.

Another area of interest is the development of dry or near dry EDM technology, which utilizes a gaseous medium as a dielectric [216–218]. A mixture of liquid and gas under pressure is proposed to be used as a dielectric in near dry EDM [218,219]. Dry EDM uses a gas medium as a dielectric [220–222]. Atmospheric air, compressed air, liquid nitrogen, oxygen, argon, and helium are the main gaseous media used in EDM [223]. Dry EDM has critical issues related to its stability and performance at the current level of technological progress [224]. At the same time, the EDM process in a gas environment is safe and promising [223]. But the idea of using gas as a dielectric medium is controversial in the scientific community due to the significantly lower erosion effect than in the dielectric liquid process. Research on the selection of dry EDM parameters for stability of the processing process is relevant [221]. Scientifically based knowledge about dry EDM is currently not sufficient for a qualitative description of the process [225]. Selection of dry EDM process parameters to ensure the necessary performance is required [223]. Currently, dry EDM can be effectively used to create small cavities or micro machining processes [226].

However, achieving true green production requires more than just changing one process component; it necessitates the careful selection of the right combination of tools, workpieces, and dielectric mediums to minimize radiation toxicity [21,215].

### 7.2. Smart Manufacturing

Smart manufacturing in EDM involves the implementation of process control and monitoring systems [23,97]. As far back as 1997, there was a proposal to use neural networks for monitoring, relying on signals related to the gap between the tool and the workpiece for various pulse types [227]. High-speed data acquisition has also been suggested for monitoring EDM pulses [228]. Additionally, the potential for real-time process control using wavelet transforms based on waveform data from voltage and current measurements between the workpiece and the electrode has been explored [229]. Real-time monitoring systems utilizing virtual sensors have been proposed [230]. Acoustic emission signals can serve as an effective mechanism for monitoring EDM processes [231]. Furthermore, it is important to track the influence of multiple factors on the technological process by incorporating monitoring systems based on vibro-acoustics [42,208,232]. While some developments related to total emission monitoring in EDM based on emission analysis exist [233], the scope of such studies remains limited due to the complexity of the process.

Various process control systems have been proposed to enhance EDM quality [23,94,234,235], including fuzzy logic-based control [23], inverse modeling [94], and adaptive control systems that monitor the spark ratio as an indicator of gap voltage [234]. Another approach involves control focused on maximizing MRR by dynamically optimizing the unloading time [235]. The advancement of smart manufacturing in EDM will see greater progress with the expansion of mathematical and computer modeling capabilities. Currently, deficiencies in modeling remain the primary constraint to the development of smart EDM.

### 7.3. New Technological Solutions, Hybrid Technologies

Hybrid processing is recognized as a crucial avenue in technology development, involving modifications to electrodes, dielectric mediums, and other process components [25,116,215]. The active development of combined and hybrid processes is driven by the need to overcome EDM's inherent limitations [22,30]. Researchers explore a range of innovations, including dry and nearly dry machine tools, machines incorporating magnetic fields or magnetic cushions, machines with ultrasonic vibrations, and those with rotating electrodes and/or workpieces [30,119,236]. Efforts to enhance process quality include introducing abrasive conductive powder into the dielectric liquid, incorporating ultrasonic vibrations, and cryogenic treatment of tool electrodes [97]. For instance, when the workpiece vibrates, processing efficiency improves, leading to enhanced material removal depth. Cryogenic-treated electrodes vibration contributes to reduced surface roughness, resulting in surface cleanliness improvements of over 20% [98].

## 8. Discussion

Since the advent of precision machining using EDM, significant progress has been made thanks to extensive research and its widespread adoption across various fields of science and technology. Advances in electronics and the fourth industrial revolution have brought substantial changes to experimental equipment. Manufacturers strive to provide innovative total solutions for manufacturing and scientific endeavors by improving the accuracy and thermostability of machine tools [237,238]. Electronica India Limited focuses on creating a comfortable work environment [238]. GF Machining Solutions (https://www.gfms.com/com/en.html (accessed on 15 November 2023)) is engaged not only in the development and production of machine tools, but also in the creation of unique software products for automation and process control with a focus on industry 4.0 [237].

There is a growing trend toward integrating EDM into the industry 5.0 framework, with a focus on minimizing its environmental impact. The main focus of scientific developments is aimed at the introduction of dry processing technology or biodegradable fluids. This fact is reviewed in detail in Sections 3 and 6. But the issue of the reuse of used machine waste is also relevant [239]. This will reduce negative environmental impacts and is supported by manufacturers such as GF Machining Solutions.

Experimental studies have yielded valuable data; however, many publications lack crucial information vital for industrial development. Often, articles fail to provide details such as the experimental setup, the recommended combination of materials used for workpiece tools and dielectrics, and the specific conditions under which the experiments were conducted. Some studies also fall short of fully describing input parameters, output results, and assessments of surface morphology and material structure. Statistics on experimental works are shown in Figure 9. Hasan et al. also reported the importance of experimental data complete descriptions for the enhancements of the technological process [20].

A notable gap exists in the availability of comprehensive studies that document long-term scientific team findings and provide practical recommendations for various aspects of the technological process, especially concerning different workpiece materials. An intriguing question arises regarding the analysis of the influence of tools and materials across a wide range of process parameters while considering external factors like temperature, air

quality, humidity, and other variables. Furthermore, there is a lack of statistical data on experimental results, and new findings often add complexity without necessarily improving the understanding of the process, leading to increased stochasticity.

To ensure efficiency and sustainability in EDM, a comprehensive analysis of the entire EDM process life cycle is essential, yet this remains largely unexplored.

Over the past decade, mathematical and numerical modeling of the EDM process has advanced significantly. Thermal analysis of the process has been well-documented, with effective models that provide insights into the thermos-mechanics of EDM. However, challenges persist in organizing feedback from the experimental industry, which can play a significant role in model development. Modeling erosion of both the workpiece and the tool is in a phase of substantial growth. There have been limited attempts to mathematically describe and numerically simulate the EDM process while analyzing the impact of tool wear on workpiece processing, including deviations from the desired shape and surface morphology.

Papazoglou et al., Gostimirovic et al., Mullya et al., and other authors also came to conclusions and reported the limitations of existing numerical models and methods for technological process numerical simulation, especially in terms of tool wear [94,188,200].

The problems of EDM numerical simulation are largely related to the stochastic nature of the process and the lack of statistics and completeness of experimental data. Another factor influencing the development of EDM models is complex chemical, physical-mechanical, and thermomechanical properties of workpiece and tool materials. To develop fundamental data on the physics and mechanics of the process necessary to create effective mathematical models and numerical algorithms, it is worth considering the EDM process not only from the technological side, but also from the scientific point of view. Tlili et al. revealed the importance of understanding and describing the process physics for optimizing both the experimental component and for creating effective numerical models [86].

When reviewing the current state of the EDM industry, it becomes evident that there are several unaddressed research areas. There is a critical need to compile and generalize the data on process parameters accumulated by the renowned scientific group.

*The Significant Results Identified during the Review*

The interesting facts about the EDM technological process can be identified from the publications:

1.  The dependence of the black layer size on the pulse duty factor is observed when processing steel with a copper-tungsten electrode; the composition of the black layer is multi-element (the main chemical elements are carbon, silicon, iron, and oxygen) [85].
2.  Classical TE can no longer meet the ever-increasing demands in EDM. Modification of the creation methods and geometric features of the TE is required [140,146]. A hollow porous electrode allows for increasing the efficiency of using the working fluid by ~5 times; coating the electrode with a heat- and corrosion-resistant layer with increased strength allows for improving processing parameters by 10–25%, etc.
3.  Modification of EDM machines using various hybrid technologies is necessary to improve the final processing result [75,236]. The use of magnetism solves the short circuit problem, electrode vibration reduces the crater and effectively dissipates discharge energy, etc. But the use of magnetism or vibration further complicates the technology's description.
4.  Taking into account the effect of corona discharging is required in models describing EDM with a curved geometry of the electrode working surface. This can be conducted using an electric model [200].
5.  Present EDM modeling approaches cannot describe the actual roughness and surface morphology after machining due to the limitations of the models in terms of describing pores formation and built-up zones near the crater [41].

### 9. Conclusions and Development Directions

The current state of research in the field of EDM is discussed in this article. The material includes an analysis of existing developments from experiment to simulation, from standard applications to innovative ones, etc. The main factors hindering the development of technology and numerical analogues of the EDM process were identified as part of this review. These include the mathematical description complexity of the process physics, the lack of data on tool wear, the difficulty of taking into account the influence of tool wear on the workpiece, etc. The lack of research results generalization is also one of the problems. The creation of technological databases documenting EDM for a wide range of workpiece materials and tools is current. The technological process results on simple steels with well-known physical, mechanical, and metallurgical properties are worth considering for understanding a number of phenomena arising within the EDM framework.

We have identified several relevant areas of research when reviewing the current state of the EDM industry:

1. The need to generalize the data on the process, accumulated by the scientific group for more than 10 years of work, in a scientific monograph.
2. Analysis of the EDM process using a workpiece and tool made from ordinary steel with a well-known and verified set of chemical, physical-mechanical, and thermomechanical properties.
3. These areas of research are a priority for further development of the technological process, as well as mathematical and numerical methods for its description.

The statistics on experimental data presented in Part 3 were carried out according to the following articles: [11,12,36,37,42,46,49,50,58,60,70,71,75,77,80,81,83,85,86,90–92,98,103–105,107, 108,110,115,117–119,121,122,124–142,146,166–168,170,171,183,188,190,197,198,201–203,206,209, 210,212–214,224]. The statistics on EDM numerical modeling data presented in Part 6 were carried out according to the following articles: [10,18,41,48,54,58,62,76,79–84,86,89,91,93,94, 110,115,133,135,144,146,168–170,186–188,190,191,195–199,201–213,240].

**Author Contributions:** Conceptualization, K.R.M., T.R.A., S.S.S. and A.A.K.; visualization, A.A.K., E.S.S., Y.S.K., S.S.S. and A.M.; writing—original draft preparation, K.R.M., T.R.A., A.A.K., E.S.S., S.S.S. and Y.S.K.; writing—review and editing, K.R.M., T.R.A., A.A.K., E.S.S., Y.S.K., S.S.S. and A.M.; funding acquisition, A.A.K. All authors have read and agreed to the published version of the manuscript.

**Funding:** This study was funded by Perm National Research Polytechnic University in the framework of the Federal Academic Leadership Program «Priority-2030».

**Data Availability Statement:** Not applicable.

**Conflicts of Interest:** The authors declare no conflict of interest.

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
