# Peer review of "Recent Trends and Developments in the Electrical Discharge Machining Industry: A Review"

_jmmp, doi:10.3390/jmmp7060204_

Round 1

Reviewer 1 Report

Comments and Suggestions for Authors

To make the paper / review more useful for readers I would recommend to concentrate more on the technological aspects of the results of review, not mainly on statistics (i.e.: choose the most interesting / important results among analysed and describe them shortly).

It would be also useful to add more information on the application of gaseous media for the EDM process. 

Comments on the Quality of English Language

I do not have major issues according to the language of the paper. Please perform proof reading in order to correct minor spelling mistakes. 

Author Response

We analyzed sources of information and added specific data. All comments have been corrected.

With best regards!

Reviewer 2 Report

Comments and Suggestions for Authors

This review provides a detailed summary of recent EDM research in term of processing mechanism, technology configuration, parameter optimization, and model simulation, and predicts priority directions for future development. Some inspiring suggestions for future research are made, such as generalizing the numerous experimental data in recent years to improve the simulation modeling; and using simple materials with known properties to study unclear phenomena in EDM processing. However, the following issues were noticed during reading:

1.     The effects of many process parameters on machining performance were reported in the paper. However, the electrical parameters, especially the discharge pulse power supply, as an important influencing factor of EDM, have been less discussed. Further studies on the effect of different electrical parameters in machining can be enriched and some researches on the functions and performance of discharge power supplies can be introduced.

2.     Part 3 summarizes the proportion of different material in the various study but gives no indication of what literature was counted. It would be beneficial for other researchers to find the literature more easily if the relevant literature was summarized (using a table for instance). The same goes for summary of researches on the input and output parameters of process later in Part 3.

3.     Part 7 describes PM-EDM research in surface processing and biocompatibility. Lines 705 to 784 summarize many studies related to optimizing the performance of the PM-EDM process, but the reports of these studies are somewhat scattered and lacking clues and logic. It would be better to categorize them more finely according to certain criteria, such as the controlled input process parameters or the main output parameters.

4.     Both 2.2 and 3 describe the influence of different process elements on the performance of machining, with respect to electrical parameters, materials and treatments of tools, workpieces, and dielectrics, etc. However, the different emphases of these two sections are not obvious and there are intertwined elements. It could be possible to synthesize the two parts into one part or to emphasize the different focuses of the two parts.

5.     Part 7, which focuses on the impact of different process characteristics on the performance of the PM-EDM process, is placed rather abruptly at the end of the text, after the introduction of the future direction of EDM development. Considering that the PM-EDM is also part of the EDM process performance study, would it be more appropriate to place it after the third part?

Comments on the Quality of English Language

There are also some format issues noticed such as acronyms (MRR and TWR, etc. appear many times in full after acronyms are introduced).

Author Response

We analyzed sources of information and added specific data.

Thank you for your detailed analysis of the article.

We have updated our material taking into account your recommendations for each item.

All comments have been corrected.

With best regards!

Reviewer 3 Report

Comments and Suggestions for Authors

1.  After each terminology firstly appear in this paper, it necessary to explain. For example, ANOVA, ET,…

2. Confirm abbreviation wording. For example, material removal rate ratio (MRR) [line 57]

3. One suggestion, “non-standard”, and “Not standard”, and “microelectrical discharge machining” and “Micro electrical discharge machining”, and “WEDM” “WIRE-EDM” as well as “electroerosion” and “electro-erosion” were used in this paper. Which is right?

4.      It is suggested that Figure 3 can add various EDM characteristics and their reference applications, rather than just explaining the meaning of proper nouns.

5. It is recommended that Figure 4 can mark the recast layer, heat-affected zone and liquid metal in more detail.

6. Please confirm the sum of the results of evaluation of publications in Figure 8 (a) and (b).

7. It is recommended to increase the resolution of Figure 6.

Comments on the Quality of English Language

Minor editing of English language required

Author Response

Thank you for your detailed analysis of the article.

We analyzed sources of information and added specific data.

We have updated our material taking into account your recommendations for each item.

 All comments have been corrected.

With best regards!

Round 2

Reviewer 2 Report

Comments and Suggestions for Authors

Thanks for your revised version, i think it can be published for the revised version

Author Response

Dear Reviewer!

It is pleasure to work with you!

Thank you!

Reviewer 3 Report

Comments and Suggestions for Authors

1.  Modified “using various ET [114].” to “using various ETs [114].” [line 285]

Comments on the Quality of English Language

Minor editing of English language required

Author Response

(The authors gave the same response as above.)
